# Extracellular Vesicles in Alzheimer’s and Parkinson’s Disease: Small Entities with Large Consequences

**DOI:** 10.3390/cells9112485

**Published:** 2020-11-15

**Authors:** Charysse Vandendriessche, Arnout Bruggeman, Caroline Van Cauwenberghe, Roosmarijn E. Vandenbroucke

**Affiliations:** 1VIB Center for Inflammation Research, VIB, 9052 Ghent, Belgium; charysse.vandendriessche@irc.VIB-UGent.be (C.V.); arnout.bruggeman@irc.VIB-UGent.be (A.B.); caroline.vancauwenberghe@irc.VIB-UGent.be (C.V.C.); 2Department of Biomedical Molecular Biology, Ghent University, 9000 Ghent, Belgium; 3Department of Neurology, Ghent University Hospital, 9000 Ghent, Belgium

**Keywords:** extracellular vesicles, Alzheimer’s disease, Parkinson’s disease, biomarkers

## Abstract

Alzheimer’s disease (AD) and Parkinson’s disease (PD) are incurable, devastating neurodegenerative disorders characterized by the formation and spreading of protein aggregates throughout the brain. Although the exact spreading mechanism is not completely understood, extracellular vesicles (EVs) have been proposed as potential contributors. Indeed, EVs have emerged as potential carriers of disease-associated proteins and are therefore thought to play an important role in disease progression, although some beneficial functions have also been attributed to them. EVs can be isolated from a variety of sources, including biofluids, and the analysis of their content can provide a snapshot of ongoing pathological changes in the brain. This underlines their potential as biomarker candidates which is of specific relevance in AD and PD where symptoms only arise after considerable and irreversible neuronal damage has already occurred. In this review, we discuss the known beneficial and detrimental functions of EVs in AD and PD and we highlight their promising potential to be used as biomarkers in both diseases.

## 1. Introduction

Alzheimer’s disease (AD) and Parkinson’s disease (PD) are the two most common neurodegenerative disorders characterized by the misfolding, aggregation and widespread accumulation of natively unfolded proteins [1,2]. In the case of AD, amyloid beta (Aβ) deposits into extracellular Aβ plaques, whereas the hyperphosphorylation of the protein Tau leads to the formation of neurofibrillary tangles inside neurons [3]. These lesions lead to the loss of synapses and neurodegeneration which result in the symptoms associated with AD [4]. In PD, the protein alpha-synuclein (α-syn) aggregates into intraneuronal Lewy bodies and Lewy neurites [5]. The second neuropathological hallmark of PD is the loss of dopaminergic neurons in the substantia nigra *pars compacta*, leading to functional impairment of the nigrostriatal pathway and the occurrence of the cardinal motor symptoms [6]. Interestingly, the Tau and α-syn pathology spreads throughout the brain in a predictable, characteristic pattern [7,8]. This is thought to be the result of their cell-to-cell transmission in a prion-like fashion, whereby the pathologic aggregates from the donor neuron induce the misfolding of their endogenous counterparts in the acceptor neuron [9]. In contrast to Tau pathology, amyloid plaques display a less predictable spreading pattern whereby their presence does not correlate well with cognitive decline either [10,11]. Consequently, the research focus has shifted towards soluble Aβ oligomers (AβOs) as being the main neurotoxic Aβ conformation capable of inducing synaptic dysfunction [11,12]. These oligomers also appear to be the most potent conformation related to the proposed hypothesis of prion-like spreading of Aβ pathology [13]. Although the exact spreading mechanism of these disease-related proteins is not completely understood, extracellular vesicles (EVs) have been proposed as one of the contributors [14].

EVs are nanosized membrane vesicles that can be classified as exosomes or microvesicles based on their mode of biogenesis. They are important mediators of intercellular communication due to their capacity to transfer proteins, lipids and genetic material to both adjacent and distant cells whereby they influence various physiological and pathological functions [15]. Although a major part of the research cited in this review utilizes specific nomenclature to describe EV subtypes, we decided to consistently use the general term “EVs”. The rationale behind this is that it is currently still very difficult to discriminate between EV subtypes due to the lack of subtype-specific markers [16]. In this review, we first discuss the role of EVs in AD. Hereby, we elaborate on how the proteolytic cleavage of the amyloid precursor protein (APP) can lead to the release of Aβ in association with EVs whereafter we review the known beneficial and detrimental functions of EVs in AD. Next, we discuss the role of EVs in the pathogenesis of PD. Finally, we focus on how these EVs provide exciting opportunities to be used as biomarkers in both diseases.

## 2. The Role of EVs in AD

### 2.1. The Convergence between the Generation of Aβ and EVs

Aβ is released by proteolytic processing of APP which can occur in a non-amyloidogenic or amyloidogenic fashion [17,18,19]. The difference between both pathways can be attributed to the initial cleavage step by either α-secretase or β-secretase, respectively. The former cleavage occurs within the Aβ domain and thereby prevents the formation of a full-length Aβ peptide. Subsequently, the large *N*-terminal ectodomain, called soluble APP α (sAPPα), is released, whereas the α-*C*-terminal fragment (α-CTF or C83) remains bound to the membrane. Alternatively, the cleavage by the β-secretase, called β-site APP cleaving enzyme-1 (BACE1), results in the formation of sAPPβ and β-CTF that consists of 99 amino acids (C99). The next cleavages occur within the transmembrane domain of α-CTF or β-CTF and are mediated through the γ-secretase complex. This leads to the release of the cytoplasmatic APP intracellular domain (AICD) and either P3 (Aβ_17–40/42_) in the non-amyloidogenic pathway or a longer Aβ species (including but not limited to Aβ_1–38/40/42_) in the amyloidogenic pathway [20]. The difference in length of the secreted Aβ peptide is attributed to the different positions where the sequential γ-secretase cleavages can occur, with Aβ_1–40_ being the most prevalent product that is generated. However, in ~10% of the cases, Aβ_1–42_ is formed, which is considered to be more directly related to AD pathology since it is more hydrophobic, aggregation prone and the major constituent of amyloid plaques [12,21].

Both APP and the secretases traffic through multiple cellular compartments and co-localize at several locations, although the exact processes are not completely unraveled. However, the following routes leading to non-amyloidogenic processing at the plasma membrane and amyloidogenic processing in intracellular compartments are generally proposed [18,19,22,23] (Figure 1). After its synthesis in the endoplasmic reticulum (ER), APP is transported through the Golgi compartment to the plasma membrane, where it is rapidly cleaved by α-secretase or re-internalized in endosomes. Alternatively, APP can directly be transported from the *trans*-Golgi network (TGN) to endosomes. These endosomes can recycle to the plasma membrane, undergo retrograde transport to the TGN or mature into late endosomes which can fuse with lysosomes for degradation. Additionally, the endosomes provide an acidic environment which is known to be ideal for BACE1 activity, hence leading to the amyloidogenic processing of APP. The active γ-secretase complex is thought to be principally localized to the plasma membrane and in the endosomal/lysosomal compartments, complementing the activity of α- and β-secretase, respectively, at these locations [18]. Little is known about how the intracellularly generated Aβ peptides are released into the extracellular compartment [23]. One option involves post-Golgi network transport pathways to the plasma membrane, but also the secretion of Aβ via EVs has gained attention [23]. In this context, it is interesting to know that late endosomes, which are also called multivesicular bodies (MVBs), can have multiple destinations: they can either fuse with the lysosome, resulting in degradation, or they can fuse with the plasma membrane, thereby releasing their intraluminal vesicles (ILVs) as EVs into the extracellular environment [15].

Several studies suggest detrimental effects of altering the trafficking of APP in AD pathogenesis with a central role for MVBs. Experiments in several APP-expressing cell lines and primary cortical neurons indicate that the sorting of APP to ILVs in a subset of MVBs occurs via the endosomal sorting complexes required for transport (ESCRT) pathway [24,25] and is dependent on the ubiquitination of APP [24]. Furthermore, the trafficking of APP into EVs has also been reported to be dependent on an Alix-Syntenin-1 pathway, independent of the ESCRT machinery [26]. In general, MVBs formed by the ESCRT-dependent pathway are destined for lysosomal degradation [15]. Indeed, the direct (i.e., via Hrs and Tsg101, both key players in the ESCRT pathway) and indirect (i.e., via Vps34, the main source of phosphatidylinositol-3-phosphate (PI3P)—see below) inhibition of the ESCRT pathway results in accumulation of APP in enlarged endosomes [24], whereas it reduces the lysosomal delivery of APP, leading to intracellular Aβ accumulation [25]. The reported resulting effect on secreted Aβ-levels is contradictory, namely increased [24] or decreased [25,27,28]. Interestingly, the levels of PI3P are decreased in affected brain regions of late-onset AD patients and several transgenic mice models of AD [24]. One of the functions of PI3P is its involvement in the recruitment of ESCRT complexes that mediate the intraluminal sorting of ubiquitinated cargo proteins in MVBs. Therefore, it can be speculated that reduced PI3P levels cause a decreased degradation of APP and its metabolites in lysosomes, leading to intracellular Aβ accumulation and/or EV secretion as an alternative cellular route for the MVBs. Furthermore, the intracellular Aβ accumulation might be enhanced due to an increased residence time of APP in endosomal compartments which harbor favorable conditions for amyloidogenic processing. In line with these hypotheses, PI3P depletion in primary mouse cortical neurons results in the intracellular accumulation of APP-CTFs due to decreased lysosomal degradation [28]. This is associated with a sphingolipid-dependent release of APP-CTF-containing EVs, which is suggested by the authors to be a way of disposing of undigested material [28]. Similarly, compromising the function of the lysosome in H4 glioblastoma cells expressing a pathogenic PS1 mutation significantly increases both the release of EVs and the concentration of Aβ associated with EVs [29].

The presence of several products of the APP metabolism in MVBs and EVs is reported by other studies as well. APP and APP-CTFs have been detected in MVBs in cultured human cerebrovascular cells [30] and in cortical neurons of APP_751SL_/PS1_M146L_ mice [31], respectively. In APP-expressing neuroblastoma SH-SY5Y cells, both APP itself, APP-CTFs and AICD are present in MVBs and on secreted EVs that were isolated from the culture medium [32]. In neuroblastoma N2a cells expressing the Swedish mutant form of APP (APP_Swe_), a minor fraction of the Aβ peptides present in MVBs is released in association with EVs [33]. Another study showed that EVs isolated from these APP_Swe_-expressing neuroblastoma N2a cells contain APP, α-CTF, β-CTF and Aβ_1–40/42_. In contrast, β-CTF and Aβ_1–40/42_ were respectively absent and lowered in EVs isolated from APP_WT_-expressing cells, whereas no difference in the total amount of secreted EVs was detected [34]. Strikingly, the APP and/or CTFs inside the EVs could be processed by γ-secretase of recipient cells, signifying that EVs can indeed act as vehicles for the intercellular transport of APP and its metabolites [34]. APP and β-CTF were detected in EVs secreted by APP_Swe_-expressing HEK293 and neuroblastoma SH-SY5Y cells as well [26]. Additionally, in vivo experiments show that EVs isolated from the hippocampus of AAV-C99-expressing mice are enriched in α-CTFs and β-CTFs compared to controls. This is even more pronounced upon treatment with a γ-secretase inhibitor, which also leads to the presence of higher molecular weight CTF species inside the EVs [35]. Furthermore, EVs isolated from brain lysates of Tg2576 mice were shown to contain APP, APP-CTFs and Aβ. The levels of APP and APP-CTFs inside these EVs are higher compared to the levels in EVs isolated from non-transgenic littermates. The APP-CTFs are enriched in EVs compared to cell lysates, but no differences in the total amount of EVs were reported [36].

MVBs also appear to play an important role in the final cleavage step of the APP proteolysis pathway that leads to the release of Aβ peptides. This cleavage is mediated by the γ-secretase complex which consists of four subunits including presenilin (PS1 or PS2, encoded by *PSEN1* and *PSEN2*). It has been shown that depending on the PS subunit, the subcellular localization of the γ-secretase complex differs [37]. If the γ-secretase complex contains PS2, it localizes specifically to late endosomes/MVBs and lysosomes whereas, in the case of PS1, it is more broadly distributed throughout the cell, including the plasma membrane. Strikingly, the γ-secretase cleavage in late endosomes/MVBs and lysosomes generates an intracellular pool of longer Aβ species. Familial AD (FAD) mutations in *PSEN2* increase the intracellular Aβ_1–42_/Aβ_1–40_ ratio even more dramatically. Moreover, FAD mutations in *PSEN1* change its subcellular localization towards that of PS2, similarly promoting the generation of intracellular Aβ_1–42_ species [37]. Interestingly, even prior to the formation of amyloid plaques, Aβ_1–42_ was shown to accumulate in intraneuronal MVBs of Tg2576 mice and this accumulation increased which age [38]. This suggests that intracellular Aβ accumulation is a very early event in the pathogenesis of AD. Other studies reported the presence of Aβ in intraneuronal MVBs in Tg2576 mice, both in vivo [25,38] and in vitro [39,40], in APPxPS1 mice [31] and in the human AD brain [38,41]. Moreover, the aggregation of Aβ peptides has been shown to occur in acidic cellular compartments, including late endosomes, inside neurons [42,43]. This indicates that MVBs are important cellular compartments for both the generation, aggregation and accumulation of Aβ peptides. Additionally, intracellular Aβ accumulation inside MVBs has been shown to affect the MVB sorting pathway as well [40]. Although the sequence and the interplay between the described events remain elusive, they suggest a feedback loop between aberrant subcellular trafficking of APP, intracellular Aβ accumulation and a dysfunction in the endolysosomal–autophagic system, the latter known to be an early pathological feature of AD [44]. Importantly, MVBs appear to be a central intracellular compartment at the intersection of these events, harboring Aβ and other products of the APP metabolism and providing a secretory mechanism for these components under the form of EVs. Interestingly, BACE1, a disintegrin and metalloprotease 10 (ADAM10) and nicastrin, a component of the γ-secretase complex, were shown to be present in EVs isolated from brain lysates of Tg2576 mice, indicating that the metabolic processing of APP could also occur here [36]. The latter was confirmed in a later study, where it was shown that APP present in brain-derived EVs isolated from Tg2576 mice and incubated in vitro in the absence of cells is cleaved to generate APP-CTFs and AICD [45]. Similarly, EVs isolated from APP-expressing Chinese hamster ovary (CHO) cells contain APP, APP-CTFs, Aβ and the secretases BACE, ADAM10, PS1 and PS2, although nicastrin could not be detected in this study [46]. Altogether, it is clear that EVs are linked with the Aβ-generating pathway that plays a central role in the pathogenesis of AD. In the next part of this review, we will discuss both the beneficial and detrimental roles that have been attributed to EVs in AD.

### 2.2. The Beneficial Roles of EVs in AD

In wild-type (WT) rats, synaptic disruption caused by the intracerebroventricular (icv) infusion of Aβ-containing human AD brain extracts was abrogated by EVs derived from N2a neuroblastoma cells or cerebrospinal fluid (CSF) from healthy donors. This protective effect was obtained when the EVs were either pre-infused or pre-incubated with the extracts and is likely due to the sequestration of Aβ assemblies on the surface of the EVs [47]. Along these lines, another study reported that N2a-derived EVs promote Aβ amyloidogenesis of both synthetic and endogenous Aβ, resulting in the formation of Aβ assemblies associated with EVs which can be more efficiently taken up and degraded by microglia in vitro [48]. According to the authors, the rationale behind this is that EVs accelerate the formation of non-toxic amyloid fibrils, thereby resulting in a decline of the formation of toxic oligomeric species [48]. Similarly, EVs derived from N2a cells [49] and astrocytes [50] were shown to catalyse the aggregation of Aβ in vitro [49,50] and in vivo [50]. However, in these reports, this resulted in a reduced uptake by N2a cells [49], accompanied by a reduced toxicity, and primary mixed glial cultures [50]. This might be explained by the notion that Aβ fibrils are less efficiently taken up by target cells than AβO [51], whereby intracellular Aβ can have pathological consequences and appears to play an important role in AD pathogenesis [52,53]. However, as a consequence, this might implicate a role for EVs in the formation of extracellular Aβ plaques. Mechanistically, the interaction between Aβ and the surface of the EVs was shown to be mediated by glycosphingolipids [48,54,55,56] and the cellular prion protein (PrP^C^) [49] present on the EV membrane. Furthermore, it was shown that the hippocampal administration of EVs isolated from N2a cells [54] or mouse primary neurons [55] in APP_SweInd_ mice causes a reduction in total Aβ levels and the amount of thioflavin-S positive plaques, which was attributed to an EV-associated Aβ uptake by microglia [54]. It is, however, important to note that the administered EVs are derived from healthy sources and that to acquire these positive effects, normally functioning microglia need to be present. The same research group reported that an oral treatment with plant-derived ceramides in APP_SweInd_ mice increases the amount of serum and brain tissue-derived EVs that are positive for Aβ and the neuronal markers neural cell adhesion molecule 1 (NCAM-1) and L1 cell adhesion molecule (L1CAM). This suggests that the treatment increases neuronal EVs harboring Aβ for clearance in vivo. Accordingly, the treatment reduced Aβ pathology and inflammation in the brain, whereas cognitive function as assessed by the Y-maze test was improved [56]. An alternative mechanism of EV-associated clearance of Aβ was reported in a study where an increased degradation of Aβ by microglial cultures upon statin treatment was attributed to the statin-induced release of insulin-degrading enzyme (IDE)-associated EVs [57]. Similarly, altering the release of IDE-containing EVs in APP_Swe_-expressing neuroblastoma N2a affects the degradation of extracellular Aβ [58].

### 2.3. The Detrimental Roles of EVs in AD

Although some studies reported beneficial effects of EVs derived from healthy sources in the pathogenesis of AD, their role might be completely different when they are secreted by cells affected by the disease process. Initially, EVs could act as a helpful mechanism to remove accumulating toxic material from an affected cell but, paradoxically, these EVs might subsequently exert detrimental effects. In line with this, EVs were proposed to be a potential escape mechanism for cells to be able to cope with excess amounts of intracellular Tau [59,60]. Some data suggest that EV-associated Aβ could be involved in plaque formation. Indeed, the EV marker Flottilin-1 partially colocalized with Aβ_1–42_ in senile plaques of Tg2576 mice [61], whereas both Alix and Flotillin-1 are enriched in amyloid plaques of AD patients compared to age-matched controls [33,62]. Additionally, Tau- and Aβ-containing EVs derived from a variety of in vitro and in vivo sources were shown to be taken up by recipient cells in vitro and in vivo, highlighting their potential as vehicles to spread AD-related proteins. In this context, both Tau and AβO-containing EVs isolated from, respectively, the brain of PS19 Tauopathy mice [63] and AD patients [41] were taken up by neuronal cultures. Subsequently, the AβO-containing EVs either release their cargo, causing cytotoxicity, or migrate to further recipient cells [41]. The latter result was reported in other studies and for Tau as well [64,65], supporting the notion that at least part of the EVs that are taken up by the target cell can be re-secreted, thereby achieving a longer distance of action. To this end, the internalized EVs were shown to hijack the secretory endosomal pathway in the recipient neurons [65]. Interestingly, blocking the EV production, secretion or uptake in neuronal cultures reduces the neuron-to-neuron spread of AβO and the related toxicity [41]. Furthermore, Aβ-containing EVs isolated from PS1 mutant neurons and the CSF of sporadic, late-onset AD patients are neurotoxic to primary rat cortical neurons, whereby there is a highly significant correlation between the EV Aβ_1–42_ levels and the induced neurotoxicity [29]. In line with this, the in vitro neuronal toxicity of human CSF-derived EVs (defined to be microvesicles) from AD patients is significantly decreased by pre-treatment with anti-Aβ antibodies [66]. Astrocyte-derived EVs (ADEs) isolated from plasma of AD patients exert a similar neurotoxicity in primary rat cortical neurons [67]. This was shown to be due to high levels of several complement proteins in these ADEs that mediate the deposition of the membrane attack complex on the neurons, causing their membrane disruption and the activation of necroptosis [67,68]. Furthermore, the addition of microglia-derived EVs (defined to be microvesicles) to Aβ_1–42_ aggregates in vitro promoted the formation of small soluble neurotoxic Aβ_1–42_ species, thereby increasing neurotoxicity in cultured primary hippocampal neurons [66].

Tau-containing EVs isolated from the culture medium of either Tau-stimulated microglia [63] or Tau-transduced, human-derived induced pluripotent stem cells (iPSCs) [69] transfer their Tau content to hippocampal neurons after their injection into the hippocampus of WT mice. Intriguingly, besides being taken up and transferring their content, Tau-containing EVs have the potential to mediate the propagation of Tau pathology. Indeed, EVs isolated from in vitro (i.e., N2a cells expressing aggregation-prone Tau [64]) and in vivo (i.e., brain of Tau transgenic rTg4510 mice [70], CSF derived from AD patients [64,71]) sources can induce Tau aggregation in in vitro systems optimized to assess Tau seeding capacity. Moreover, hippocampal injection of EVs derived from rTg4510 brain induces Tau phosphorylation at the site of injection in Tau transgenic ALZ17 mice [72]. Similar injection setups in WT mice using EVs isolated from either FAD patient-derived [73] or Tau-transduced [74] neuronally differentiated iPSCs revealed the same results. Furthermore, neuron-derived, phosphorylated Tau (p-Tau)-containing EVs from AD and mild cognitive impairment (MCI) patients, but not cognitively normal controls, can propagate Tau pathology upon their injection into the brain of WT mice [75]. Functional studies showing beneficial effects of inhibiting EV secretion add additional weight to the hypothesis that EVs play a detrimental role in the AD disease process. Indeed, pharmacologically inhibiting EV synthesis by the systemic administration of GW4869, an inhibitor of neutral sphingomyelinase 2 (nSMase 2), slows Tau propagation in a mouse model of rapid Tau propagation [63] and reduces the plaque load in 5xFAD mice [50]. Moreover, genetically nSMase 2-deficient 5XFAD mice display a reduction in both plaque burden and Tau phosphorylation, whereas cognitive function is improved [76]. Depleting microglia or microglial Bin1 reduces the EV-associated Tau content in vivo and in vitro and supresses the propagation of Tau pathology in Tauopathy mouse models as well [63,71].

Altogether, these data clearly point towards a detrimental role for EVs in the pathogenesis of AD. However, it is important to add that only a minor part of extracellular Aβ (<1%) and Tau (<0.2% to ~2–3%) is associated with EVs [33,64,77]. In line with this, the total levels of Aβ_1–40/42_ in EVs isolated from plasma and CSF samples from AD patients were shown to be drastically lower compared to the EV-free fraction of the respective sample types [29]. Nevertheless, the Aβ_1–42_/Aβ_1–40_ ratio was significantly higher in these EVs, underlining their importance in AD pathogenesis [29]. Additionally, Tau phosphorylated at Threonine 181 (pT181) [78] and oligomeric Tau [64] were suggested to be enriched in human CSF-EVs compared to total CSF in, respectively, mild AD patients and both AD patients and controls. Moreover, EV-associated Tau could induce Tau aggregation in vitro [64] and Tau transduction in vivo [63], whereas naked Tau was unable to achieve these effects. Similar findings were reported for EV-associated Aβ since EVs isolated from PS1 mutant neurons induce neurotoxicity in primary rat cortical neurons to the same extent as EV-depleted medium, although the latter contains about 10-fold more Aβ_1–42_ compared to the EVs [29].

## 3. The Role of EVs in PD

Next to AD, EVs have also been implicated in the pathogenesis of PD which is characterized by the misfolding and aggregation of α-syn followed by its deposition in Lewy bodies and Lewy neurites. Because α-syn lacks an ER targeting sequence, it was originally considered as an exclusively intracellular protein. However, this concept was challenged by initial reports showing the presence of extracellular α-syn in human CSF [79,80] and plasma [80,81]. Interestingly, studies in human α-syn-expressing SH-SY5Y neuroblastoma cells indicated that α-syn is continuously secreted via an ER/Golgi-independent mechanism involving the endocytic pathway, whereby at least part of the secreted α-syn is associated with EVs [82,83,84]. Furthermore, it has been shown that α-syn can interact with the ESCRT pathway, which ultimately can result in its release in association with EVs [85,86]. Several studies collectively indicate that α-syn monomers, oligomers and fibrils can be located both in the lumen and on the surface of EVs [82,83,87,88,89,90,91,92]. However, similar to what has been described for Aβ and Tau, only a minor component of the extracellular α-syn associates with EVs [87,89,90,93,94,95,96]. Nonetheless, several observations point towards an important role for this fraction in the pathogenesis of PD. Intravesicular α-syn, present in human α-syn-expressing SH-SY5Y neuroblastoma cells, was shown to be more prone to aggregation compared to cytosolic α-syn [82]. EVs can also accelerate the aggregation of α-syn in vitro, whereby EVs derived from control N2a cells exert the same effect as EVs derived from α-syn-overexpressing N2a cells [89]. Of specific relevance, it was reported that EVs isolated from the CSF of patients with PD and dementia with Lewy bodies (DLB), but not control patients, induce α-syn aggregation in vitro as well [95]. The addition of EVs isolated from WT mouse brain induces the assembly of recombinant α-syn preformed fibrils (PFFs) into higher-order multimers in vitro. In the same timeframe, no multimerization of α-syn PFF in the absence of EVs was observed [97]. However, similarly to what was described for Aβ [47], the pre-incubation of α-syn PFFs with these EVs neutralized the detrimental effects of α-syn PFFs in vitro (i.e., reduced uptake by primary cortical cultures) and in vivo (i.e., inability to induce α-syn accumulation after intrastriatal injection in WT mice) [97]. Interestingly, the A53T α-syn mutant displays an increased association with EVs, leading to the speculation that pathogenic α-syn species may be preferentially sorted into EVs [90]. In line with this, misfolded and oligomeric α-syn (i.e., presumably the more toxic form of α-syn) were shown to be favorably released with EVs compared to, respectively, native and highly aggregated α-syn [84,98].

After their secretion, α-syn-containing EVs can be taken up by recipient neurons in vitro, where they induce toxicity [87,99]. Of interest, the uptake of EV-associated oligomeric α-syn species derived from α-syn-expressing H4 neuroglioma [87,94] and SH-SY5Y neuroblastoma [90] cells by the respective naive cells is more efficient compared to that of EV-free α-syn. In line with this, sonication of EVs derived from α-syn-overexpressing cell lines reduces their internalization and the transfer of α-syn to recipient cells, indicating that the EVs need to be intact to efficiently exert these functions [87,88]. A functional study investigating the effect of nSMase 2 inhibition in SH-SY5Y cells further underlined the role of EVs in the spreading of α-syn between neuronal cells. Indeed, nSMase 2 inhibition results in reduced EV secretion in response to oligomeric α-syn. This is accompanied by a reduced transfer of oligomeric α-syn to recipient cells [100]. Additionally, in vivo data show that the intrastriatal injection of EVs isolated from human α-syn-expressing HEK cells in WT mice resulted in the spreading of the human α-syn towards interconnected brain regions [99]. Similarly, α-syn neuronal inclusions were detected close to the injection site upon the intracortical administration of CSF-derived EVs from DLB patients in WT mice [92]. Strikingly, the uptake and transfer of α-syn-containing EVs has been linked to the prion-like spreading of α-syn. EVs obtained from α-syn PFFs stimulated primary cortical neurons act as seeds to induce endogenous α-syn aggregation in recipient neurons in vitro [101]. Additionally, brain-derived EVs from DLB patients facilitate α-syn aggregation in vivo after their intracerebral injection in WT mice [102]. Similar results were seen after the intrastriatal injection of plasma-derived EVs from PD patients [103]. Furthermore, α-syn-containing EVs derived from serum of PD patients trigger α-syn aggregation, dopaminergic degeneration and motor deficits when administered into the striatum of WT mice [104]. However, EVs isolated from A53T α-syn transgenic mouse brain were unable to induce α-syn pathology and motor deficits after their intrastriatal injection in WT mice [97].

Although all data described above focus on neuronal-derived EVs, there also appears to be a role for microglia-derived EVs in PD pathogenesis. EVs derived from the plasma of PD patients are internalized by microglia in vitro and in vivo, thereby inducing microglial activation in both setups. Subsequently, BV-2 microglia were shown to secrete α-syn-containing EVs which induce α-syn aggregation in SH-SY5Y recipient cells [103]. The stimulation of BV-2 microglia cells with α-syn results in the release of EVs that are able to induce apoptosis in recipient neurons [105]. Additionally, α-syn PFF-stimulated primary microglia secrete α-syn oligomer-containing EVs which trigger α-syn aggregation in vitro (i.e., in primary cortical neurons) and in vivo (i.e., upon intrastriatal injection in WT mice), leading to the loss of dopaminergic neurons and the occurrence of motor deficits in the latter situation [101]. Interestingly, microglia/macrophage-derived CD11b positive EVs are present in the CSF of PD and multiple system atrophy (MSA) patients and are able to induce α-syn aggregation in vitro [101]. Together, these data underline the potential of microglia in facilitating the propagation of α-syn via EVs.

Several PD-related assets can be linked with EV secretion. Environmental toxins, including certain pesticides and metals, are hypothesized to play a role in the initiation and progression of PD pathology. For example, chronic exposure to high doses of the neurotoxic metal manganese results in manganism which is characterized by many Parkinsonian features [106]. Interestingly, manganese enhances the release of α-syn oligomer-containing EVs from α-syn-expressing mouse dopaminergic neuronal cells. When these EVs are injected into the striatum of WT mice, they propagate α-syn pathology and induce the development of Parkinsonian motor deficits [107]. Additionally, exposure to the pesticide rotenone is associated with developing PD [108]. In this context, it was shown that rotenone is able to induce the release of α-syn in association with EVs from WT primary neurons [91]. Collectively, these studies indicate a potential role for α-syn-containing EVs connecting environmental triggers and PD. Furthermore, another molecule implicated in PD pathogenesis, namely the lipid peroxidation product 4-hydroxynonenal (HNE), induces the aggregation of endogenous α-syn in primary cortical neurons as well as the release of α-syn oligomer-containing EVs [99]. An increasing body of evidence links a dysfunction in the endolysosomal–autophagic system, which is increasingly recognized as a central event in the pathophysiology of PD, to EV secretion too [109,110,111]. Indeed, blocking the fusion of the autophagosome (AP) with the lysosome by Bafilomycin A1 (Baf) treatment in α-syn-overexpressing neuronal cell lines increases vesicular α-syn release [84,87,88,92,98]. This is associated with a higher transmission of α-syn to recipient SH-SY5Y cells [88]. Mechanistically, the Baf treatment results in the accumulation of fused AP-MVB compartments in α-syn-overexpressing H4 cells, whereby the EVs secreted by these cells contain several AP markers [92]. Furthermore, blocking autophagy by silencing of autophagy-related gene 5 (ATG5) in α-syn-overexpressing Lund human mesencephalic (LUHMES) neuronal cells protects them against the α-syn-induced toxicity by increasing the α-syn secretion in EVs. As such, the EVs provide a compensatory rescue mechanism for impaired intracellular autophagy [112]. Another association between lysosomal dysfunction and PD is provided by the significant relevance of the lysosomal enzyme glucocerebrosidase (GCase) in PD. Up to 7% of PD patients carry a mutation in the *GBA1* gene [113], whereas PD patients that do not carry a *GBA1* mutation also show a significant reduction in lysosomal GCase activity [114,115]. Interestingly, the pharmacological inhibition of GCase by conduritol-B epoxide (CBE) in A53T α-syn transgenic mice significantly increases brain EV levels and EV-associated oligomeric α-syn [116]. A *Drosophila* model of GCase deficiency shows markedly increased EV numbers in the hemolymph (i.e., the *Drosophila* equivalent of blood) [117]. Moreover, it is suggested that the excessive EV secretion caused by the GCase deficiency might be responsible for the characteristic presence of protein aggregates in this model, since knocking down components of the ESCRT pathway reduces the accumulation of these protein aggregates [117]. Strikingly, the ratio of EV α-syn to total α-syn in plasma of PD patients is inversely correlated with GCase enzymatic activity, which again suggests that GCase activity might be linked to the release of EVs [118].

## 4. EVs as Potential Biomarkers for AD and PD

AD and PD pathology begins multiple years before symptoms arise, hence the diagnosis is made when considerable and irreversible neuronal damage has already occurred [119,120]. Therefore, it is of great importance to identify patients in the earliest stages of the disease, when potential disease-modifying treatments might have the greatest benefit. To this end, biomarkers are of specific interest. Next to diagnosing and monitoring disease, they could be instrumental in evaluating the patients’ response to disease-modifying treatments. Interestingly, several studies already indicated that EV-associated proteins might provide a snapshot of the molecular events occurring in the brain of patients with neurodegenerative diseases and thereby serve as a liquid brain biopsy. For example, an analysis of brain tissue-derived EVs revealed that Aβ and p-Tau are present in EVs isolated from AD and DLB patients, whereas in the latter condition they contain α-syn as well [102]. Furthermore, EVs isolated from the temporal neocortex of AD patients contain significantly more AβO compared to EVs from non-neurological control samples [41]. These results indicate that the EV cargo can reflect brain pathology. Another study demonstrating a significant positive correlation between the elevated levels of Tau phosphorylated at Serine 396 (pS396) and Aβ_1–42_ in brain tissue homogenate and brain-derived EVs of AD patients supports this notion [121]. EV-bound Aβ derived from plasma of AD patients was shown to strongly correlate with brain plaque deposition as assessed by positron emission tomography (PET), whereas this was not the case for total plasma Aβ [122]. These results further demonstrate that brain biomarkers can be detected in the periphery as well. Similarly, several miRNAs were shown to be upregulated in both brain tissue and serum-derived EVs of AD patients, although there was little overall correlation between miRNA expression profiles in brain homogenate, brain tissue-derived EVs and serum-derived EVs [123]. However, the authors suggest that enriching for specific cell type-derived EVs from serum might improve this. Indeed, EVs that are separated from plasma or serum based on their expression of the neuronal surface marker L1CAM are enriched for neuronal proteins compared to the total EV population and control EV subpopulations [124]. Alternatively, NCAM can also be used to enrich for neural-derived EVs (NDEs). Using the latter method, it was shown that the levels of total Tau (t-Tau), pT181 Tau and Aβ_1–42_ in plasma NDEs highly correlate with their respective levels in the CSF [125]. This further underlines the potential of NDEs to reflect pathological changes in the brain [125]. A proteomics analysis of brain tissue-derived EVs revealed that half of the quantified proteins displayed neuron-specific markers and the other half glia-specific markers, whereby the former are enriched in the control group and the latter in the AD group [121]. Another study reported the presence of several central nervous system (CNS) cell type markers (i.e., neurons, microglia, oligodendrocytes and astrocytes) on human brain tissue-derived vesicles that could be explored to capture specific cell type-derived EVs in the periphery [126]. In summary, both the total population and specific cell type enriched EVs in a variety of sources have emerged as potential biomarkers due to their role in the disease process and their ability to carry disease-related molecules.

### 4.1. EVs as Biomarkers in AD

Although the majority of studies investigating EV-related biomarkers in AD have been focusing on blood as an easily accessible biofluid, some results were reported in CSF. Both EVs derived from CSF of AD patients and controls have been shown to contain Tau species that are phosphorylated at several sites (i.e., pS262/pS356, pS396/pS404, pT181) [64,78]. The amount of p-Tau in CSF-EVs did not differ between those two groups in one study [64], whereas another report described an increase in the ratio of EV p-Tau to CSF t-Tau specifically in early AD patients [78]. The number of isolectin B4 (IB4) positive particles (defined as myeloid microvesicles) is increased in the CSF of AD patients and MCI patients converting to AD in comparison with stable MCI patients and controls [127]. These levels are associated with hippocampal atrophy and white matter tract damage in AD and MCI patients, respectively [127]. A proteomics analysis on EV separated from CSF samples of AD patients, MCI patients and corresponding controls identified the heat shock 70 kDa protein 1A (HSPA1A), puromycin-sensitive aminopeptidase (NPEPPS) and prostaglandin F2 receptor negative regulator (PTGFRN) as differentially expressed proteins, whereby HSPA1A and NPEPPS are significantly increased in AD patients versus MCI patients and PTGFRN is significantly increased in AD patients versus controls [128]. Additionally, a variety of biomarker candidates have been identified in plasma- or serum-derived NDEs of AD patients. Several of these biomarkers have promising abilities to distinguish patients from controls, whereby some of them might aid disease staging and even predict disease development from a preclinical stage. These results are summarized in Table 1. As an indication of the diagnostic ability of the identified biomarkers, the area under the receiver operating curve (ROC) is included when available. Alternatively, the sensitivity, specificity or *p*-value are mentioned.

NDEs derived from plasma or serum from AD patients contain increased levels of pT181 Tau, pS396 Tau and Aβ_1–42_ compared to controls [75,125,134,143]. Strikingly, this is already the case for up to ten years before their diagnosis when these patients were still cognitively normal [134]. Moreover, the levels of those three markers are increased in MCI patients converting to AD in comparison with stable MCI patients as well [75]. Therefore, this set of biomarkers has the potential to distinguish different stages of AD whereby both persons at risk at their preclinical stage and MCI patients that are most likely to progress to AD can be identified. Additionally, the plasma NDE levels of pT181 Tau and Aβ_1–42_ highly correlate with their levels in the CSF, further underlining the potential of NDEs to reflect pathological changes in the brain [125]. A large case–control study analyzed several of these biomarker candidates in NDEs separated from longitudinal plasma samples collected prior to AD disease onset which occurred on average four years after collecting the earliest preclinical sample. NDEs from future AD patients contained longitudinally elevated mean levels of pT181 Tau and pT231 Tau in comparison with controls, whereas t-Tau and, surprisingly, Aβ_1–42_ remained unaltered. Furthermore, the NDE diameter was increased but no differences in NDE concentration could be observed. Combining several repeated measures in a prediction model could predict AD with high specificity [135]. Another study did not report differences between the pT181 Tau levels in NDEs from patients with AD, MCI and controls [136]. The plasma NDE levels of pT181 Tau and Aβ_1–42_ in AD patients also do not correlate with cognitive decline [138], whereas the mean longitudinal plasma NDE levels of pT181 Tau in preclinical AD patients were associated with cognitive performance [135]. Total Tau levels in plasma-derived NDEs were shown to be indifferent between AD patients and controls, but significantly upregulated in PD patients where they correlate with disease severity [142]. In contrast, increased levels of t-Tau have also been reported in plasma-derived NDEs from AD patients [125]. Differences in the methods to enrich for NDEs and to assess the t-Tau levels might provide an explanation for these discrepancies. Next to Aβ, other products of the APP processing pathway (i.e., sAPPα and sAPPβ) are also increased in plasma NDEs from AD patients compared to controls [143]. Serum-derived NDEs isolated from older individuals with age-related cognitive decline milder than MCI or dementia also show higher pT231 Tau, pT181 Tau and t-Tau in comparison with age-matched cognitively stable individuals [148].

Alterations in insulin signaling, such as dysregulated phosphorylation of the insulin receptor substrate-1 (IRS-1), which can trigger insulin resistance, have been linked to the pathogenesis of AD [137,149]. This is reflected in plasma-derived NDEs of AD patients since the levels of IRS-1-pS312 and IRS-1-pTyr are increased and decreased, respectively, compared to controls [137]. Moreover, these altered levels correlate with brain atrophy in the AD patients, again supporting that peripheral EV biomarkers can reflect changes in the AD brain [150]. Furthermore, the insulin resistance index (i.e., the ratio of IRS-1-pS312 to IRS-1-pTyr) is significantly increased in the AD condition. Strikingly, these differences already plateaued in preclinical patients up to ten years before their AD diagnosis was made [137]. In line with these data, the mean IRS-1-pS312 levels amongst longitudinal plasma-derived NDEs of preclinical AD patients were significantly elevated in comparison with controls and were associated with worse cognitive performance. However, in this sample set, the IRS-1-pTyr levels were significantly increased in preclinical AD patients as well [135]. Importantly, the levels of several insulin signaling mediators in serum-derived NDEs were shown to be associated with a longitudinal, prospective change in cognitive performance in a cohort of individuals with and without age-related cognitive decline [148]. The plasma NDE levels of several synaptic proteins and transcription factors mediating neuronal protection are significantly decreased in multiple AD stages (preclinical AD patients, MCI patients converting to AD, AD patients) compared to controls and stable MCI patients [75,138,139,140,141]. In contrast, the levels of multiple lysosomal proteins are significantly increased in the preclinical AD group compared to controls [139]. At the stage of diagnosis one to ten years later, the levels of almost all synaptic proteins declined even further, whereas they remained constant for most lysosomal proteins and the transcription factors [138,139,140,141]. The detailed information of all investigated biomarkers of each category can be found in Table 1. These data suggest the identification of multiple promising preclinical biomarker candidates, whereas several synaptic proteins additionally hold promise as prognostic biomarkers. Furthermore, the altered levels of several biomarkers in NDEs correlate with the Mini Mental State Examination (MMSE) score of AD patients, suggesting that they could be used to follow the progression of cognitive decline. This is the case for the synaptosomal-associated-protein 25 (SNAP25) in serum NDEs and the synaptic proteins synaptopodin, synaptophysin, GluA4-containing glutamate (AMPA4) and neuroligin 1 (NLGN1) in plasma-derived NDEs, of which the levels are decreased in AD patients compared to controls [133,138,141]. A similar positive correlation with the MMSE score was shown for the decreased levels of miR-223 in EVs isolated from serum of dementia patients, including both AD and vascular dementia cases [130]. Since plasma and serum-derived EVs have been shown to be enriched in miRNAs compared to cell-free plasma and serum, the miRNA profile of EVs has also attracted considerable attention as a potential biomarker [151]. The levels of miR-193b are lower in EVs derived from serum of AD patients compared to MCI patients, which in turn show decreased levels compared to controls [129,132]. Conversely, miR-384 show the inverse pattern, whereas miR-135a is equally increased in MCI and AD patients [132]. Both in serum- [131] and plasma- [145] derived EVs of AD patients, panels of differentially expressed miRNAs were identified that could differentiate AD patients from controls. In plasma NDEs, miR-212 and miR-132 are decreased in AD patients compared to controls [146]. Furthermore, decreased miR-100-3p levels and increased miR-23a-3p, miR-223-3p and miR-190a-5p levels were reported in plasma NDEs of AD patients in comparison with controls [147].

Next to NDEs, ADEs, isolated using an antibody against the glutamine aspartate transporter (GLAST), have also been investigated. It has been shown that the total number of plasma ADEs is significantly lower than the number of plasma NDEs in AD patients and controls. Nonetheless, the plasma ADEs contain significantly higher levels of a number of proteins (i.e., BACE1, γ-secretase, sAPPα, sAPPβ, pT181, pS396 and Aβ_1–42_) compared to NDEs [143]. From this set of proteins, the levels of BACE1 and sAPPα are significantly higher, whereas the levels of Aβ_1–42_ and the neural protein septin-8 are significantly lower in ADEs of AD patients compared to controls [143]. As a side note, the authors mention that these significant but moderate differences might be less useful to diagnose AD. However, they might provide a way to monitor the effect of, for example, drugs that aim to inhibit BACE1 [143]. Furthermore, a variety of complement proteins are significantly upregulated in plasma ADEs of AD patients at the dementia stage, but not yet in the preclinical AD phase. In contrast, the levels of the complement regulatory proteins CD59 and decay-accelerating factor (DAF) are already decreased at the preclinical phase and further decline as the disease progresses [68]. Since complement regulatory proteins normally prevent excessive complement activation, their decrease in the preclinical phase might be the cause of complement activation in astrocytes later on [68]. Another subtype of specific EVs that has been isolated are EVs derived from chondroitin sulfate proteoglycan (CSPG) 4 type neural precursor cells [144]. These EVs are less abundant in comparison with ADEs and NDEs in plasma of AD patients and controls, but they contain significantly higher levels of several neurotrophic factors whereby the levels are lower in both preclinical and later stage AD patients compared to controls [144]. However, because of the low abundance of CSPG4 EVs, their biomarker potential will likely be quite limited [144].

Interestingly, EVs have already been used for treatment follow-up in clinical trials. One trial demonstrated improved cognition after a daily subcutaneous treatment with growth hormone-releasing hormone (GHRH) over 20 weeks in MCI patients [152]. Plasma NDEs derived from those patients contain significantly higher levels of Aβ_1–42_ and significantly lower levels of several synaptic proteins (i.e., neurogranin, synaptophysin, synaptotagmin, synaptopodin) compared to controls. However, the levels of none of these biomarkers were significantly altered due to the GHRH treatment [153]. In the Study of Nasal Insulin in the Fight Against Forgetfulness (SNIFF120) clinical trial, AD and MCI patients that were treated daily with an intranasal dose of 20 international units (IU) of insulin for four months showed improved cognition [154]. This outcome correlates positively with changes in IRS-1-pS312 and IRS-1-pTyr in plasma NDEs of these patients, although no significant difference in the levels of these markers between baseline and the study endpoint were detected. In the future, these results could be helpful to determine which patients are most likely to benefit from the treatment [155].

### 4.2. EVs as Biomarkers in PD

In PD patients, the total level of EVs was reported to be unchanged [156] or lowered [157] in serum, unchanged in plasma [118] and increased in CSF [95]. Next to EV levels, a growing number of studies found several promising candidate biomarkers associated with these EVs in a variety of biological sources. These results are summarized in Table 2. In CSF samples, the α-syn EV levels are lower in early stage PD patients compared to healthy controls. In contrast, no differences could be detected in more advanced stage PD patients compared to neurological controls (i.e., polyneuropathy and progressive supranuclear palsy (PSP) patients). However, compared to the latter three groups, EVs derived from CSF of DLB patients show decreased levels of α-syn which correlate with the patients’ severity of cognitive impairment as assessed by the MMSE scores [95]. Another study specifically investigated the microglia/macrophage-derived, CD11b positive EVs in CSF samples of PD, MSA and control patients. Although the total number of EVs and the fibrillary α-syn levels do not differ, both the total and oligomeric α-syn levels are increased in PD and MSA patients [101]. In plasma, the α-syn levels in both the total pool [103,118] and the L1CAM positive [158,159] pool of EVs are significantly elevated in PD patients compared to controls. More specifically, one study reported that the total, monomeric and oligomeric α-syn levels in these plasma EVs are increased, whereas no differences were detected in fibrillar α-syn [103]. Similarly, serum EVs isolated from PD patients were shown to contain elevated levels of total, monomeric, oligomeric and pSer129 α-syn [104]. Of note, these EVs also contain higher levels of the inflammatory cytokines tumor necrosis factor alpha (TNF-α) and interleukin 1 beta (IL-1β) [104]. Furthermore, total α-syn levels in L1CAM positive serum EVs are elevated approximately two-fold in conditions with Lewy body pathology compared to MSA and unrelated neurodegenerative diseases [160]. As an important readout to monitor and predict disease progression, it was shown that α-syn in plasma L1CAM positive EVs correlates with disease severity as assessed by the Unified Parkinson’s Disease Rating Scale (UPDRS) score [158]. However, another report did not identify this association [159]. Additionally, no correlation between α-syn in serum-derived, L1CAM positive EVs and the UPDRS or Montreal cognitive assessment (MoCa) score was found [160]. In the total EV pool in plasma of PD patients, a negative correlation was found between the ratio of EV α-syn to plasma α-syn and disease severity, as assessed by the Hoehn and Yahr (H&Y) score [118]. 

Next to α-syn, other EV-related proteins that could also serve as biomarker candidates were identified. DJ-1 is significantly elevated in EVs isolated from plasma of PD patients compared to controls [159]. Additionally, certain miRNAs are more (i.e., miR-24 and miR-195) or less (i.e., miR-19b) abundant in serum EVs of PD patients [166]. In plasma-derived EVs of PD patients, miR-331-5p is increased, whereas miR-505 is decreased [162]. Several mitochondrial markers (i.e., ATP5A, NDUFS3 and SDHB) are decreased in serum EVs of PD patients as well [157,165]. More generally, a Raman spectroscopy analysis revealed that the molecular fingerprint of EVs isolated from the serum of PD patients is different compared to controls. Therefore, this technique might hold promise for the diagnosis and monitoring of PD once further optimized and validated [163]. Proteomics analyses from serum-isolated EVs revealed that several proteins are differentially abundant between PD patients and controls [156,161,164]. For example, EVs from the plasma of PD patients contain less clusterin, complement C1r and apolipoprotein A1 (ApoA-1) compared to controls [161]. Of particular interest, certain proteins are significantly altered between different PD stages as well, underlining their potential as biomarkers to trace the progression of PD [161,164]. For example, ApoA-1 is significantly lower in plasma EVs of H&Y stage III compared to stage II PD patients [161]. Additionally, in serum-derived EVs, a proteomics approach identified seven proteins that are progressively upregulated from mild to severe PD, whereas seven other proteins progressively decreased [164]. These progressively upregulated proteins include clusterin and complement C1r, which contradicts the results that were obtained in plasma-derived EVs [161,164]. Next to potential prognostic biomarkers, some promising results suggesting the use of EV biomarkers to distinguish PD from other disorders were reported. In CSF, both EV-associated α-syn and the ratio of EV α-syn to the total number of EVs could discriminate between PD, dementia with Lewy bodies and neurological controls (i.e., polyneuropathy and progressive supranuclear palsy patients) with relatively high sensitivity and specificity [95]. Furthermore, the combination of α-syn and clusterin in serum NDEs allows us to discriminate between PD and both MSA (AUC: 0.91) and other proteinopathies (i.e., frontotemporal dementia, corticobasal syndrome and PSP) (AUC: 0.98) [160].

Not only are CSF and plasma useful sources of potential EV-associated biomarker candidates, since interesting results were obtained in the easily accessible biofluids saliva and urine as well. The number of EVs in saliva from PD patients is increased compared to controls [173]. Additionally, saliva-derived EVs from PD patients contain higher levels of oligomeric α-syn, whereas the ratio of total α-syn to oligomeric α-syn is also increased compared to controls [172]. However, no association with disease severity was found [172]. In urinary EVs, SNAP23 and calbindin are elevated in PD patients compared to controls [171]. Similar results were obtained for DJ-1, although this only holds true for samples derived from males [168]. In contrast, no differences in leucine-rich repeat kinase 2 (LRRK2) were present [168], which was reported in other studies as well [167,169]. Mutations in *LRKK2*, which can be phosphorylated at multiple sites, are one of the most commonly known genetic causes of PD [174]. Interestingly, the ratio of Ser(P)-1292 LRRK2 to total LRKK2 is increased in urinary EVs of PD patients with and without *LRKK2* mutations, whereby the Ser(P)-1292 LRRK2 levels in the latter group correlate with several non-motor aspects of PD [169,170].

One study already reported the use of EVs for treatment follow-up in a clinical trial, namely the Exenatide-PD trial. In this trial, the treatment group demonstrated a slight improvement in their UPDRS motor score, whereas the control group continued to worsen [175]. Concomitantly, the levels of IRS-1-pTyr increased in serum-derived NDEs from the Exenatide group compared to the placebo group, suggesting an increase in brain insulin signaling pathways due to the treatment. Additionally, an increase in downstream effector molecules (i.e., total and phosphorylated mechanistic target of rapamycin (mTOR)) in these serum NDEs could be correlated with the improved UPDRS motor score in the treatment group [176]. These data support the usability of EVs as tools to validate a treatment response.

## 5. Conclusions

CSF is the only biofluid that is currently included in the guidelines for the biomarker-based definition of AD provided by the research framework of the National Institute on Aging–Alzheimer’s Association (NIA-AAA) and the International Working Group (IWG) [177,178]. The CSF biomarker profile defining AD consists of a low CSF Aβ_1–42_ or Aβ_1–42_/Aβ_1-40_ ratio, elevated CSF p-Tau and elevated CSF t-Tau [177,178]. This CSF biomarker profile is not standardly used in clinical practice but can increase diagnostic accuracy when the clinical diagnosis is inconclusive [179]. For PD, no validated biomarker profile is available yet. PD diagnosis is based on clinical symptoms as described in the International Parkinson and Movement Disorder Society Clinical Diagnostic Criteria for PD [180]. Therefore, there is a great need for novel and additional biomarkers with the ability to predict, diagnose and follow-up disease. Preferentially, these biomarkers should be derived from easily accessible biofluids. Although the collection of CSF via a lumbar puncture is a safe procedure [181], it still remains rather invasive. However, the alternative use of blood-based candidate biomarkers implicates several challenges. Blood is a complex fluid in which potential biomarkers are susceptible to potential proteolytic degradation and in which they are generally present at a low concentration [182,183]. The usability of α-syn as a reliable biomarker in blood is further complicated because the majority of α-syn that is present in blood derives from red blood cells [184]. EVs can overcome some of these challenges and therefore provide a highly interesting alternative biomarker source. Because of their ability to cross the brain barriers [185], they can be detected in peripheral biofluids in which they protect their cargo from degradation. Furthermore, the biomarker of interest is often enriched in EVs in comparison with the total biofluid. This can improve the measurement sensitivity, lower the detection threshold and enhance the signal-to-noise ratio. The use of specific markers to enrich for EVs derived from cellular sources that are known to be involved in the disease process even further improves these advantages [124]. Nonetheless, the use of EVs as biomarkers also comes with several considerations. A variety of isolation methods are currently available to separate EVs from biological fluids, as is reflected in Table 1 and Table 2. Each method has its own specificity and efficiency, thereby influencing the identification of EV cargo [186]. Quality control measures are of utmost importance to verify the depletion of contaminants and the successful separation of EVs from a given biofluid [16]. Additionally, pre-analytical variables in collecting (e.g., timepoint of sampling), processing (e.g., centrifugation speed) and storing (e.g., temperature) samples prior to the EV isolation can influence the obtained results [187,188]. Punctiliously reporting all experimental parameters and quality control measurements as proposed by the transparent reporting and centralizing knowledge in EV research (EV-TRACK) initiative and the minimal information for studies of EVs (MISEV) guidelines is therefore indispensable for reproducible and reliable EV research [16,186]. Prior to the potential implementation of EV assays in the clinic, fully validated and standardized protocols will be needed [189]. Furthermore, the identified EV-related biomarker candidates will need to be validated in large datasets to verify the initial results. Although there is still a long way to go, EV biomarker research is an exciting field which holds great promise to be of value to patients in the future.

## Figures and Tables

**Figure 1 cells-09-02485-f001:**
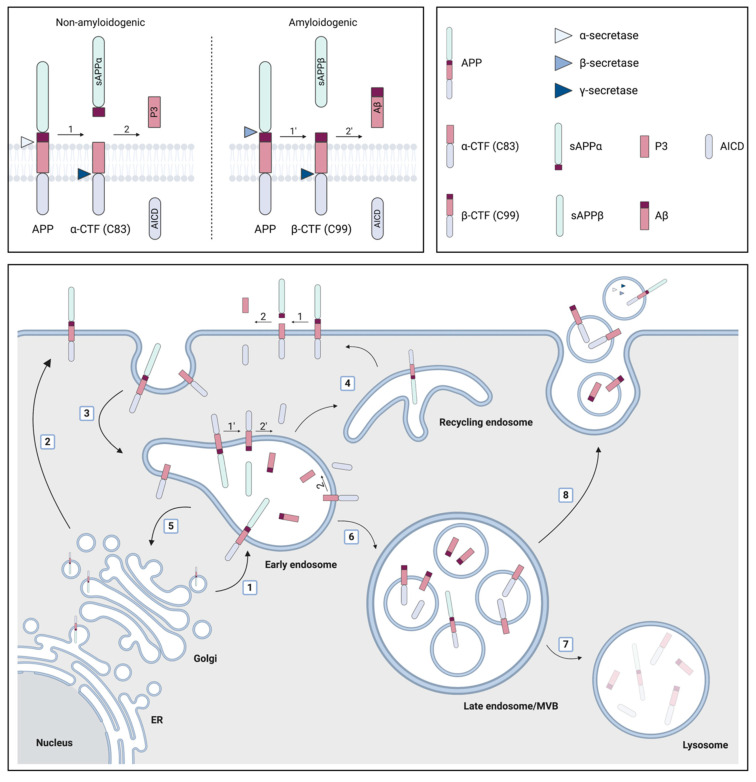
The role of multivesicular bodies (MVBs) and extracellular vesicles (EVs) in amyloid precursor protein (APP) processing. APP is synthesized in the endoplasmic reticulum (ER) and subsequently transported through the Golgi compartment to early endosomes (1) or the plasma membrane (2), where it can be cleaved by α-secretase in the non-amyloidogenic pathway. This cleavage leads to the formation of soluble APP α (sAPPα) and the α-C-terminal fragment (α-CTF or C83). Alternatively, APP can be re-internalized in endosomes (3) where the amyloidogenic cleavage by β-secretase preferentially occurs. Thereby, sAPPβ and β-CTF (C99) are generated. The formation of the APP intracellular domain (AICD) and either the P3 fragment or the amyloid β (Aβ) peptide from α- and β-CTF, respectively, requires cleavage by the γ-secretase complex. Early endosomes recycle to the plasma membrane (4), undergo retrograde transport to the trans-Golgi network (5) or mature into late endosomes/MVBs (6). The latter either fuse with lysosomes (7) or with the plasma membrane (8), resulting in the extracellular release of their intraluminal vesicles (ILVs) as EVs. Several products of the APP processing pathway have been localized to MVBs (i.e., APP itself; α- and β- CTF; AICD; Aβ) and EVs (i.e., APP itself; α- and β- CTF; Aβ; α-, β- and components of the γ-secretase complex), whereby Aβ can be localized both on the inside and on the surface of EVs. Figure created with BioRender.

**Table 1 cells-09-02485-t001:** Overview of extracellular vesicle (EV)-associated biomarkers in Alzheimer’s disease (AD).

			EV QC					AD vs. ctrl			
Source	Cohort	EV Isolation Method	NTA	TEM	EV Markers	Category	Biomarker	Result	Statistics	Additional Potential	Ref
CSF	AD: 10 Ctrl: 10	Differential centrifugation	x	x	+	Neurotoxic protein	p-Tau	=	ns		[64]
	AD: 21 Ctrl: 9	Ultracentrifugation + sucrose gradient fractionation			+		Ratio EV pT181 Tau/EV t-Tau	↑	*p* = 0.04	Prognostic	[78]
							Ratio EV pT181 Tau/CSF pT181 Tau		*p* = 0.002		
	AD: 7 ctrl: 7	Total Exosome isolation kit				miRNA	miR-193b	↓	*p* < 0.05		[129]
Serum	AD: 22 Ctrl: 16	ExoQuick	x		+		miR-223	↓	AUC = 0.875		[130]
	AD: 39 MCI: 11 Ctrl: 59	Plasma/Serum exosome isolation kit					miRNA signature (16 miRNAs)	≠	Sens: 87% Spec: 77%		[131]
	AD: 51 MCI: 43 ctrl: ?	Total Exosome isolation kit					miR-193b	↓	*p* < 0.05	Prognostic	[129]
	AD: 107 MCI: 101 Ctrl: ?	Total Exosome isolation kit	x		+		miR-135a	↑	Sens: 95% Spec: 96%		[132]
							miR-384		Sens: 97% Spec: 99%	Prognostic	
							miR-193b	↓	Sens: 94% Spec: 86%		
	AD: 25 Ctrl: 17	ExoQuick + immunoprecipitation with L1CAM			+	Synaptic protein	SNAP25	↓	AUC = 0.826		[133]
Plasma/serum	AD: 57 Ctrl: 57 FU AD: 24	ExoQuick + immunoprecipitation with L1CAM	x		+	Neurotoxic proteins	t-Tau	=			[134]
							pS396 Tau pT181 Tau	↑	AUC = 0.999	Preclinical	
							Aβ1-42			Preclinical, prognostic	
Plasma	AD 1: 120 Ctrl 1: 222 AD 2: 35 Ctrl 2: 29		x	x	+; -	Neurotoxic proteins	Aβ1-42	=	Optimal model cohort 1: AUC^train^ = 0.896; AUC^test^ = 0.8 Optimal model cohort 2: AUC^train^ = 0.989; AUC^test^ = 0.767		[135]
							t-Tau				
							pT181 Tau	↑		Preclinical	
							pT231				
						Insulin signaling	IRS-1-pS312				
							IRS-1-pTyr				
	AD: 10 ADC: 20 MCI: 20 Ctrl: 10		x	x	+	Neurotoxic proteins	pT181 Tau	↑	AUC = 1	Prognostic	[75]
							pS396 Tau		AUC = 0.98		
							Aβ_1–42_		AUC = 0.98		
						Synaptic protein	Neurogranin	↓	AUC = 1		
						Survival factor	REST		AUC = 1		
	AD: 20 MCI: 10 Ctrl: 10		x		+	Neurotoxic proteins	t-Tau pT181 Tau	=	ns		[136]
	AD: 26 Ctrl: 26 FU AD: 22				+	Insulin signaling	IRS-1-pS312	↑	AUC = 0.932	Preclinical	[137]
							IRS-1-pTyr	↓	AUC = 1		
							Ratio IRS-1-pS312/IRS-1 pTyr	↑	AUC = 1		
	AD: 12 Ctrl: 12 FU AD: 9				+	Synaptic protein	Synaptotagmin	↓	AUC = 0.99	Preclinical Prognostic	[138]
							Synaptophysin		AUC = 1		
							Synaptopodin		AUC = 0.97		
							Neurogranin		AUC = 0.99		
							GAP43		AUC = 0.79		
	AD: 26 Ctrl: 16 FU AD: 20				+	Lysosomal proteins	LAMP-1	↑	*p* = 0.00051	Preclinical	[139]
							Cathepsin D		AUC = 1		
							Ubiquitinylated proteins				
						Heat-shock protein	HSP70	↓			
	AD: 24 Ctrl: 24 FU AD: 16				+	Survival factors	LRP6	↓	AUC = 0.924	Preclinical	[140]
							HSF1		AUC = 0.944		
							REST		AUC = 0.944		
	AD: 28 Ctrl: 28 FU AD: 18				+	Synaptic proteins	AMPA4	↓	*p* < 0.0001	Preclinical Prognostic	[141]
							NLGN1				
							NRXN2				
							NPTX2		*p* < 0.01		
	AD: 106 Ctrl: 106	Immunocapture with L1CAM	x			Neurotoxic protein	Tau	=	ns		[142]
	AD: 12 Ctrl: 10	ExoQuick + immunoprecipitation with L1CAM or GLAST	x		+	Neurotoxic proteins	BACE1	ADE ↑ NDE =	*p* < 0.0001; ns		[143]
							Gamma secretase	ADE = NDE =	ns; ns		
							sAPPα	ADE = NDE ↑	ns; *p* = 0.0008		
							sAPPβ	ADE ↑ NDE ↑	*p* = 0.0159; *p* = 0.0028		
							pT181 and pS396 Tau	ADE = NDE ↑	ns; -		
							Aβ1-42	ADE ↓ NDE ↑	*p* < 0.05; -		
						Neural protein	Septin-8	ADE ↓ NDE =	*p* < 0.0001; ns		
	AD: 28 Ctrl: 28 FU AD: 16	ExoQuick + immunoprecipitation with GLAST	x	x	+; -	Inflammatory cytokines	IL-6, TNF-α, IL-1β	↑	*p* < 0.001; *p* < 0.01; *p* < 0.001		[67,68]
						Complement proteins	C1q, C3b, C3d, C4b, C5b-C9 TCC		*p* < 0.0001		
							Factor B, Factor D, Fragment Bb				
							CR1, CD46	↓	*p* < 0.01		
							CD59, DAF		*p* < 0.0001	Preclinical, prognostic	
	AD: 24 Ctrl: 24 FU AD: 15	ExoQuick + immunoprecipitation with CSPG4 and PDGFRa	x		+	Neurotrophic factors	HGF	↓	*p* < 0.0001	Preclinical	[144]
							FGF2				
							IGF1				
							FGF13		*p* < 0.01		
	AD: 101 MCI: 96 Ctrl: 101	ExoQuick + immunoprecipitation with NCAM		x	+	Neurotoxic proteins	t-Tau	↑	AUC^train^ = 0.87; AUC^test^ = 0.89	Preclinical Prognostic	[125]
							pT181 Tau		AUC^train^ = 0.89; AUC^test^ = 0.88		
							Aβ1-42		AUC^train^ and AUC^test^ = 0.93		
	AD: 35 Ctrl: 35	Differential centrifugation			+	miRNA	miRNA signature	≠	AUC = 0.919		[145]
	AD: 31 MCI: 16 Ctrl: 16	ExoQuick + immunoprecipitation with L1CAM			+		miR-212 miR-132	↓	AUC: 0.77 AUC: 0.84		[146]
	AD: 40 Ctrl: 40		x	x	+		miR-100-3p miR-23a-3p miR-223-3p miR-190a-5p	↓	*p* = 0.008		[147]
								↑	*p* = 0.008 *p* = 0.016 *p* = 0.003		

This table only discusses results obtained in Alzheimer’s disease (AD) patients, patients with mild cognitive impairment (MCI) and controls (ctrl). Follow-up (FU) AD patients provided samples at two time-points: first when cognitively intact and later when diagnosed with AD. ADC patients are MCI patients who are converting to AD. For reference [135], cohort 1 consists of longitudinal samples collected from AD patients prior to symptom onset and respective controls, whereas cohort 2 consists of longitudinal samples of clinical AD patients and respective controls. In the extracellular vesicle (EV) markers column, the + symbol indicates the assessment of typical EV markers whereas the—symbol indicates the assessment of non-EV markers. For statistics, preferably the area under the receiver operating curve (AUC) or alternatively the sensitivity (sens) and specificity (spec) or *p*-value are indicated. Preclinical biomarkers have the potential to identify patients at an early disease stage before the occurrence of clinical symptoms. Prognostic biomarkers have the potential to differentiate between different stages of AD and follow-up disease progression. Abbreviations: QC, quality control; NTA, nanoparticle tracking analysis; TEM: transmission electron microscopy; CSF, cerebrospinal fluid; p-Tau, phosphorylated Tau; t-Tau, total Tau; SNAP25, synaptosomal-associated-protein 25; Aβ, amyloid beta; REST, repressor element 1-silencing transcription factor; IRS-1, insulin receptor substrate-1; GAP43, growth-associated protein 43; LAMP-1, lysosome-associated membrane protein 1; HSP70, heat-shock protein 70; LRP6, lipoprotein receptor-related protein 6; HSF1, heat-shock factor-1; AMPA4, GluA4-containing glutamate; NLGN1, neuroligin 1; NRXN2, neurexin 2a; NPTX2, neuronal pentraxin 2; BACE1, β-site APP cleaving enzyme-1; sAPPα, soluble APPα; sAPPβ, soluble APPβ; IL-6, interleukin 6; TNF-α, tumor necrosis factor α; IL-1β, interleukin 1 β; TCC, terminal complement complex; CR1, complement receptor type 1; DAF, decay-accelerating factor; HGF, hepatocyte growth factor; FGF, fibroblast growth factor; IGF1, insulin-like growth factor 1; L1CAM, L1 cell adhesion molecule; NCAM, neural cell adhesion molecule; GLAST, glutamine aspartate transporter; CSPG4, chondroitin sulfate proteoglycan 4; PDGFRα, platelet growth factor receptor α; ADE, astrocyte-derived EV; NDE, neural-derived EV.

**Table 2 cells-09-02485-t002:** Overview of extracellular vesicle (EV)-associated biomarkers in Parkinson’s disease (PD).

			EV QC				PD vs. ctrl		
Source	Cohort	EV Isolation Method	NTA	TEM	EV Markers	Biomarker	Result	Statistics	Reference
CSF	PD: 76 Ctrl: 58	Differential centrifugation	x	x	+; -	α-syn	↓	*p* < 0.05	[95]
	PD: 9 Ctrl: 9	CD11b immunocapture				α-syn	↑	*p* < 0.05	[101]
						Oligomeric α-syn			
						Fibrillar α-syn	=	ns	
Plasma	PD: 267 Ctrl: 215	L1CAM immunocapture		x	+	α-syn Ratio EV α-syn/plasma α-syn	↑	AUC: 0.654	[158]
								AUC: 0.657	
	PD: 39 Ctrl: 33	Differential centrifugation	x	x	+; -	α-syn Ratio EV α-syn/plasma α-syn	↑	*p* < 0.001	[118]
	PD: 20 Ctrl: 15	Differential centrifugation		x	+; -	α-syn	↑	*p* = 0.02	[103]
						Monomeric and oligomeric α-syn		*p* = 0.0002; *p* < 0.0001	
						Fibrillar α-syn	=	ns	
	PD: 39 Ctrl: 40	ExoQuick + immunoprecipitation with L1CAM		x		α-syn	↑	AUC: 0.654	[159]
						DJ-1		AUC: 0.703	
						Ratio EV DJ-1/plasma DJ-1		AUC: 0.724	
	PD: 16 Ctrl: 8	Size exclusion chromatography		x	+	Clusterin	↓	*p* < 0.05	[161]
						Complement C1r			
						Apolipoprotein A1			
	PD: 52 Ctrl: 48	PureExo exosome isolation kit		x	+	miR-331-5p	↑	AUC: 0.849	[162]
						miR-505	↓	AUC: 0.898	
Serum	PD: 14 Ctrl: 14	Differential centrifugation	x	x	+; -	α-syn	↑	*p* < 0.05	[104]
						Monomeric and oligomeric α-syn		-	
						pS129 α-syn			
						TNF-α and IL-1β		*p* < 0.05	
	PD: 230 Ctrl: 144	Differential centrifugation + immunoprecipitation with L1CAM	x		+	α-syn	↑	AUC: 0.86	[160]
						Clusterin	=	ns	
	PD: 22 Ctrl: 18	Size exclusion chromatography		x	+; -	Raman spectrum	≠	AUC: 0.71	[163]
	PD: 36 Ctrl: 36	Differential centrifugation	x	x	+; -	Protein composition	≠	NA	[156]
	PD: 20 Ctrl: 10	Differential centrifugation				Protein composition	≠	NA	[164]
	PD: 16 Ctrl: 12	Differential centrifugation		x	+; -	ATP5A	↓	*p* < 0.0001	[157,165]
						NDUFS3			
						SDHB			
	PD: 109 Ctrl: 40	Total exosome isolation reagent			+	miR-24	↑	AUC: 0.908	[166]
						miR-195		AUC: 0.697	
						miR-19b	↓	AUC: 0.753	
Urine	PD: 20 Ctrl: 15	Differential centrifugation		x	+	LRKK2	=	ns	[167]
	PD: 26 Ctrl: 21	Microfiltration			+	DJ-1	↑ (only in males)	*p* = 0.0493	[168]
						LRKK2	=	ns	
	LRKK2+ PD: 21 LRKK2- PD: 20 LRKK2+ noPD: 16	Differential centrifugation			+	Ratio pSer1292 LRKK2/total LRKK2	LRKK2+ PD/LRKK2- PD: ↑	AUC: 1	[169]
							LRKK2+ PD/LRKK2+ noPD: ↑	AUC: 0.844	
	PD: 79 Ctrl: 79	Differential centrifugation		x	+	pSer1292 LRKK2	↑	*p* = 0.0014	[170]
	PD: 28 Ctrl: 22	Differential centrifugation	x	x		SNAP23	↑	AUC: 0.8	[171]
						Calbindin		AUC: 0.75	
Saliva	PD: 74 Ctrl: 60	XYCQ EV enrichment kit	x	x	+	α-syn	=	ns	[172]
						pS129 α-syn			
						Oligomeric α-syn	↑	AUC: 0.941	
						Ratio α-syn/oligomeric α-syn		AUC: 0.772	
	PD: 18 Ctrl: 5	PEG precipitation	x		+	α-syn	↑	*p* < 0.0004	[173]
						L1CAM		*p* < 0.0001	

This table only discusses results obtained in Parkinson’s disease (PD) patients and controls (ctrl). In the extracellular vesicle (EV) markers column, the + symbol indicates the assessment of typical EV markers whereas the—symbol indicates the assessment of non-EV markers. For statistics, preferably the area under the receiver operating curve (AUC) or alternatively the sensitivity (sens) and specificity (spec) or *p*-value are indicated. Abbreviations: QC, quality control; NTA, nanoparticle tracking analysis; TEM, transmission electron microscopy; CSF, cerebrospinal fluid; L1CAM, L1 cell adhesion molecule; α-syn, alpha synuclein; TNF-α, tumor necrosis factor α; IL-1β, interleukin 1 β; NA, not applicable; ATP5A, adenosine triphosphate 5A; NDUFS3, NADH:ubiquinone oxidoreductase subunit S3; SDHB, succinate dehydrogenase complex iron sulfur subunit B; LRKK2, leucine-rich repeat kinase 2; SNAP23, synaptosomal-associated protein 23.

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
