# Peer review of "Extracellular Vesicles in Alzheimer’s and Parkinson’s Disease: Small Entities with Large Consequences"

_cells, 2020, doi:10.3390/cells9112485_

Round 1

Reviewer 1 Report

this is a comprehensive review about the role of small EVs in Alzheimer's disease and Parkinson disease.

the authors nicely approached and reviewed the topic. The use of figures and tables help the reading. The structure of the review is correct and fluid and the language is fine.

Some minor points should be addressed as a careful check of the contraction of some specific words es ADEs dos not have the extension.

Moreover some recent papers are not listed,as for instance the one of Serpente et al. Cells. 2020 Jun 10;9(6):1443. doi: 10.3390/cells9061443. PMID: 32531989; PMCID: PMC7349735.

Author Response

This is a comprehensive review about the role of small EVs in Alzheimer's disease and Parkinson disease. The authors nicely approached and reviewed the topic. The use of figures and tables help the reading. The structure of the review is correct and fluid and the language is fine.

We thank the referee for these positive comments!

Some minor points should be addressed as a careful check of the contraction of some specific words es ADEs dos not have the extension.

We re-checked the manuscript carefully to ensure that all abbreviations were introduced the first time they were used, which was already the case for ADEs on line 249.

We added the abbreviation for ADAM 10 (line 183), CNS (line 423), TNF-α (line 669) and IL-1β (line 669).

Moreover some recent papers are not listed, as for instance the one of Serpente et al. Cells. 2020 Jun 10;9(6):1443. doi: 10.3390/cells9061443. PMID: 32531989; PMCID: PMC7349735.

The reference was added to the manuscript (line 614) and included in Table 1 (line 487).

Additional changes to the manuscript

To further improve the quality of the manuscript, the following changes were included as well:

  • Figure 1: font size is increased to ameliorate readability
  • Table 1 and 2: in the quality control column, it is now indicated whether also non-EV markers were investigated (shown by symbol: -) next to assessing the presence of EV markers (shown by symbol: +).
  • Line 304: sentence was slightly adapted to avoid misinterpretation

Changes to the text (tracked with track changes)

  • Line 16, Line 29, Line 31, Line 45, Line 140, Line 183, Line 304, Line 423, Line 529, Line 614, Line 669, and Line 725
  • Figure 1
  • Table 1 and Table 2

Reviewer 2 Report

The review article by Vandendriessche C, et.al. entitled “Extracellular vesicles in Alzheimer’s and Parkinson’s disease: small entities with large consequences” discussed the role of exosome as biomarker in Alzheimer’s and Parkinson disease. Authors have discussed the hallmark of both the neurological disorder in detailed and the role of Exosomes in the progression of the disease.

The MS is provides in-depth knowledge about AD mainly, but given less space for PD (may be because of a lesser amount of available data). A short paragraph on PD progression can be added.  The MS is organized and nicely written.  Some of the minor comments must be incorporate to improve the quality of MS.

Minor Comments are:

  1. Please rewrite line no 16..”However, also some…
  2. Please remove “interestingly” from line no 17.
  3. Line no 29. Please included the consequences of tau hyperphosphorylation and plaque formation..Like synaptic impairment etc.
  4. Please make a new para from line no 40..from EVs introduction
  5. Please look at line 131 (APP751SLPS1M146L is there is space in between).
  6. Authors can have a look on following MS.

https://pubmed.ncbi.nlm.nih.gov/30699987/

https://pubmed.ncbi.nlm.nih.gov/29033828/

Author Response

The review article by Vandendriessche C, et.al. entitled “Extracellular vesicles in Alzheimer’s and Parkinson’s disease: small entities with large consequences” discussed the role of exosome as biomarker in Alzheimer’s and Parkinson disease. Authors have discussed the hallmark of both the neurological disorder in detailed and the role of Exosomes in the progression of the disease.

We thank the referee for these positive comments!

The MS is provides in-depth knowledge about AD mainly, but given less space for PD (may be because of a lesser amount of available data). A short paragraph on PD progression can be added.  The MS is organized and nicely written.  Some of the minor comments must be incorporate to improve the quality of MS.

We included the following sentence in the manuscript (line 31):

The second neuropathological hallmark of PD is the loss of dopaminergic neurons in the substantia nigra pars compacta, leading to functional impairment of the nigrostriatal pathway and the occurrence of the cardinal motor symptoms.

We believe that a more elaborate introduction on PD progression is outside the scope of the current review, as this was also not included for AD. It is true however that the part on AD and EVs is more extensive compared to PD, since there is a part on how the Aβ biogenesis is linked to EVs and this is not applicable on PD.

Minor Comments are:

1. Please rewrite line no 16..”However, also some…

We adapted the sentence to:

Indeed, EVs have emerged as potential carriers of disease-associated proteins and are therefore though to play an important role in disease progression although also some beneficial functions have been attributed to them.

2. Please remove “interestingly” from line no 17.

 We adapted the sentence accordingly.

3. Line no 29. Please included the consequences of tau hyperphosphorylation and plaque formation..Like synaptic impairment etc.

We included the following sentence (line 29):

These lesions lead to the loss of synapses and neurodegeneration which results in the symptoms associated with AD.

4. Please make a new para from line no 40.from EVs introduction

 A new paragraph was added.

5. Please look at line 131 (APP751SLPS1M146L is there is space in between).

We adapted the spelling to APP751SL/PS1M146L (line 140).

6. Authors can have a look on following MS.

While these reviews are interesting, we hope that the reviewer understands that we decided not to add them to the current MS as there are many more reviews of interest that could potentially be added as well.

Additional changes to the manuscript

To further improve the quality of the manuscript, the following changes were included as well:

  • Figure 1: font size is increased to ameliorate readability
  • Table 1 and 2: in the quality control column, it is now indicated whether also non-EV markers were investigated (shown by symbol: -) next to assessing the presence of EV markers (shown by symbol: +).
  • Line 304: sentence was slightly adapted to avoid misinterpretation

Changes to the text (tracked with track changes)

  • Line 16, Line 29, Line 31, Line 45, Line 140, Line 183, Line 304, Line 423, Line 529, Line 614, Line 669, and Line 725
  • Figure 1
  • Table 1 and Table 2